# Responses of Pigs to Stunning with Nitrogen Filled High-Expansion Foam

**DOI:** 10.3390/ani10122210

**Published:** 2020-11-25

**Authors:** Cecilia Lindahl, Erik Sindhøj, Rebecka Brattlund Hellgren, Charlotte Berg, Anna Wallenbeck

**Affiliations:** 1Department of Agriculture and Food, RISE Research Institutes of Sweden AB, 750-07 Uppsala, Sweden; Erik.Sindhoj@ri.se; 2Department of Animal Environment and Health, Swedish University of Agricultural Sciences, 532-23 Skara, Sweden; rehn0002@stud.slu.se (R.B.H.); Lotta.Berg@slu.se (C.B.); Anna.Wallenbeck@slu.se (A.W.)

**Keywords:** animal welfare, hypoxia, anoxia, aversion, controlled atmosphere, killing, euthanasia, gas

## Abstract

**Simple Summary:**

Stunning pigs with carbon dioxide gas is one of the most common methods for commercial slaughter. Carbon dioxide, however, has been shown to be aversive for pigs and causes a high degree of distress before they lose consciousness. Stunning with nitrogen gas is less aversive than with carbon dioxide, and an innovative method that delivers the nitrogen gas in high-expansion foam in a closed container could potentially improve pig welfare at stunning. Pigs were exposed to either air-filled foam, nitrogen-filled foam, or no foam in air, and the behavioural and physiological responses were assessed. The pigs did not show any strong aversive behaviours when exposed to foam, regardless of whether it was air-filled or nitrogen-filled foam. However, they seemed to avoid putting their heads and snouts into the foam, and the rate of escape attempts through the lid increased when foam levels became high. Five minutes after the nitrogen foam production started, the pigs were assessed to be in deep unconsciousness or dead. Based on the results found, stunning with the nitrogen foam technique may be a viable alternative to carbon dioxide stunning and offer improved animal welfare. Further studies are needed to assess the new method for stunning of slaughter-weight pigs.

**Abstract:**

Nitrogen gas (N_2_) delivered in high expansion foam in a closed container could be a feasible method for humanely stunning pigs. This study aimed to evaluate potential aversion in pigs to the N_2_ foam method and its effect on stun quality. Furthermore, the study aimed to assess potential aversion to the foam itself. Sixty pigs (27.8 ± 4.4 kg) were divided into three treatments and were exposed to either N_2_-filled foam, air-filled foam, or no foam air. The N_2_ foam was effective at purging the air from the container and quickly created stable anoxic conditions. The pigs did not show any strong aversive behaviours when exposed to foam. However, they seemed to avoid putting their heads and snouts into the foam when foam levels became high. Escape attempts through the lid also increased when the foam started covering their heads. The mean time to loss of posture was 57.9 s. Based on the results, stunning with the N_2_ foam technique could be a viable alternative to high concentration CO_2_ stunning and potentially lead to improved animal welfare at slaughter.

## 1. Introduction

Proper stunning of production animals at slaughter is under scrutiny from public authorities and consumers. In Europe, high-concentration carbon dioxide (CO_2_) stunning is a commonly used stunning method for pigs due to efficiency and assumed benefits for animal welfare and meat quality compared to electrical stunning [1,2]. CO_2_ stunning allows for pigs to be handled and stunned in small groups instead of individually, with minimized human-animal contact and reduced stress related to separation from conspecifics [3,4]. However, the high concentration of CO_2_ gas has been shown to induce aversion in pigs prior to loss of consciousness [2,3,5]. CO_2_ gas at high concentration is acidic when inhaled and can cause painful irritation to nasal mucosa [6] and has been shown to cause air hunger and breathlessness, which may be a sign of severe distress [3]. The European Food Safety Authority (EFSA) [7] has defined animal-based measures for pigs related to pain, fear and respiratory distress during exposure to high concentrations of CO_2_ to be associated with excitation behaviour, retreat, escape attempts, and gasping. In 2004, EFSA concluded that CO_2_ stunning is not optimal from an animal welfare perspective and that further research is needed to find less aversive or non-aversive gas mixture alternatives. A recent review has shown that only limited research on stunning alternatives has been published since then and that a viable alternative is still lacking [8].

For EU member states, permitted ways of stunning are described in European Council Regulation 1099/2009 regarding the welfare of animals at the time of killing. Under this regulation, stunning with inert gases like nitrogen and argon is permitted for pigs. Nitrogen gas (N_2_) has been suggested as a more humane alternative than CO_2_ for stunning pigs, as the gas in itself is not painful to inhale and studies on the behavioural response of pigs to N_2_ have indicated that pigs show less aversion compared to CO_2_ [9,10,11]. However, for inert gas stunning to be effective, it requires quickly achieving and maintaining a stable anoxic atmosphere (<2% volume oxygen) [3]. This seems difficult to achieve with N_2_ gas, as it has a slightly lower density than air and is difficult to retain. Only one published study was found that used high-concentration N_2_ gas to stun pigs, which demonstrated difficulty establishing and maintaining a stable anoxic atmosphere [12]. Very few research studies have been published on high-concentration N_2_ stunning in pigs, most likely due to these technical challenges related to quickly producing and maintaining a sufficiently anoxic atmosphere with N_2_ gas.

The use of high-expansion foam filled with N_2_ in a closed container presents a potentially feasible method to ensure a rapid elimination of oxygen and create a sustained anoxic atmosphere. When filling the container, the foam effectively purges air and thereby eliminates exposure of animals to a prolonged period of slowly decreasing oxygen levels when gas without foam would be mixing with the air as it is flushed out. The N_2_ foam technology, like CO_2_ stunning, could also offer advantages like group stunning and minimal human–animal handling. However, the foam itself may cause stress, and during the period the animal is covered in foam, it is not possible to visually observe the stunning process. One study on poultry showed that euthanasia achieved with N_2_ high-expansion foam was effective and rapid with only a few aversive responses [13]. Limited research has been conducted on the effects of this method on animal welfare in pigs, but it is nevertheless approved in some European countries for on-farm euthanasia of piglets. A doctoral thesis by Pöhlmann [14] assessed the stunning of slaughter-weight pigs using an early version of the N_2_ foam technology in an open container, with results indicating animal welfare issues related to stun quality, consistency, and duration. The results of Pöhlmann [14] need to be evaluated in relation to the study’s limitations both regarding the methodology and readiness of the technique, which has since then been developed further. The technique’s capability to quickly provide and sustain an anoxic atmosphere with high-concentration N_2_ gas makes it interesting for stunning purposes, but there is a need for a scientific evaluation of the method from an animal welfare perspective.

The aims of this study were to assess the behavioural and physiological responses of pigs to high-concentration N_2_ gas delivered in a high-expansion foam and to evaluate the stun quality. Furthermore, the study aimed to assess potential aversion to the foam itself by including an air-filled foam treatment.

## 2. Materials and Methods

### 2.1. The Stun System and Test Equipment

The stun box (110 × 92 × 67 cm) used was designed and made by the Dutch company Anoxia B.V. (Figure 1). The box was equipped with two flat, high-capacity foam generators mounted on one side of the box and a gas jet pulse system along the base to break the bubbles so the animals could be observed. Two 50-litre bottles with either compressed N_2_ or air (200 bar; AirLiquide gas AB, Uppsala, Sweden), reduced to 7 bar per bottle, were connected to the control unit of the stun box. A solution of water and 3% foam agent was used for foam production. The box had a side door to allow pigs to walk into the box and a lid that could open to ease access. Both the floor and the lid had transparent Plexiglass windows to enable video recordings from above and from below. The bottom of the box was taped with clear adhesive anti-slip tape to prevent pigs from slipping. Coloured tape was used to visually divide the floor into four equally sized quadrants (See Figure 1).

The box had two oxygen sensors to monitor conditions in the box during operation. One was a galvanic sensor (Greisinger GOX 20, Regenstauf, Germany), which was not moisture-sensitive but lacked data logging capabilities. The other sensor was a flow-through fluorescence-based electrode obtained from SST-Sensing (UK) put into an assembly with a sampling pump controlled by a microcomputer and data logged to an SD card. This sensor was moisture-sensitive and was turned off from when the foam reached it to when the foam had been destroyed.

The pigs were video-recorded in the stun box using two cameras. One video camera (Panasonic HV-X920) was attached to a stand above the box and filmed through the lid, and one (GoPro Hero 3+) was placed in a stand in the culvert under the box and filmed through the floor. A microphone, connected to the upper camera, was placed inside the box. Digital clocks were taped to the lid and floor of the box and placed so they were visible on the video recordings to facilitate synchronization of the videos. Heart and respiratory rate were recorded using a Zephyr BioModule and BioHarness attached around the pig’s chest with an elastic band. Electrode gel (Cefar Blågel, Malmö, Sweden) was used to improve contact between the BioHarness electrodes and the pig’s skin.

### 2.2. Pigs and Housing

Ethical approval for animal experiments was received from the ethical committee in Uppsala, Sweden (ref.no. 5.8.18-09852/2018). The experiments were conducted at the research facility Swedish Livestock Research Centre, Swedish University of Agricultural Sciences, Uppsala, Sweden. Sixty pigs of approx. nine weeks of age, from six different litters, were used in the experiments. The pigs were crossbreeds of Yorkshire×Dutch Yorkshire dams and Hampshire sires.

The piglets were born at the research facility. They were weaned at five weeks of age and remained in the farrowing pens for an additional five weeks after weaning. Feed and water were available ad libitum. The pigs were weighed at nine weeks of age, three days prior to the start of the experiments (Table 1).

The sixty pigs were divided into three treatments (Control, Air foam and Nitrogen foam), with 20 pigs per treatment. Pigs within pen and sex were divided into groups of three, and within these groups, one pig was randomly selected for each treatment. Thus, each treatment included 9 immuno-castrated males and 11 females from the same pens.

### 2.3. Experimental Procedure

The experiments were conducted during four consecutive days in March 2019. All pigs from one pen were moved to a temporary pen, where they were allowed to acclimatise for at least 15 min. One pig at a time was moved to a holding pen, and the BioHarness was attached (Figure 2). The pig acclimatised for two minutes before being moved to the stun box (Figure 3 and Figure 4) and for another two minutes inside the stun box before the treatment started. The pigs in the stun box were video-recorded, and heart and respiratory rates were recorded with the BioHarness. Between each test pig, the box was rinsed clean with water.

The three treatments were as follows:Control: The pig was in the box for an additional 5 min after the acclimatisation period. Oxygen levels were logged during a limited time period.Air foam: The foam production was started after the acclimatisation period and turned off when the box was filled with air foam. After ten seconds, the foam was destructed with a gas pulse and oxygen levels were logged. The treatment was terminated five minutes after the start of foam production.Nitrogen foam: The foam production was started after the acclimatisation period and turned off when the box was filled with N_2_ foam. After ten seconds, the foam was destructed with a gas pulse and oxygen levels were logged. Five minutes after the start of foam production, the box was opened, and the pig was pulled out of the box. The stun quality was assessed using a protocol (Table 2), and a stethoscope was used to control for heartbeats before sticking and bleeding.

### 2.4. Behavioural Observations

The behaviour of the pigs was assessed according to the definitions listed in Table 2 by one observer watching the videos from both above and below the stun box. It was not possible to blind the observer to the treatments that each pig was subjected to due to the innate differences between the treatments. The video recordings from below were mainly used, as this angle provided the best overview of the body, snout position and the movements of the pig, even when covered in foam. The video recordings from above were used to score vocalisations and to observe where escape attempts were directed, when exploring of the lid occurred and for other behaviours when not visible from below, e.g., when the view was obscured by foam.

In the Air foam and Nitrogen foam treatments, behaviours were continuously observed for a total of two minutes, starting 30 s before the foam production started. For Controls, behaviours were continuously recorded for a period equivalent to the foam treatments, i.e., the last 30 s of the acclimatisation period and an additional 90 s period. Behaviours were registered in 10 s intervals, resulting in 12 intervals where the first 3 intervals were pre-treatment. For the pigs in the Nitrogen foam treatment, the time from the start of the foam production until loss of posture (LOP) was recorded. The time to loss of posture, defined by the inability of the pig to remain in a standing position, was considered the first indicator of the onset of unconsciousness [6]. Convulsions (muscular excitation) were described qualitatively by intensity and type of convulsions, e.g., kicking, gagging, and the time until last observed muscular contraction was registered.

### 2.5. Stun Quality Assessment

Five minutes after the start of the N_2_ foam production, the pigs were taken out of the stun box, and the stun quality was assessed before bleeding. The stun quality was controlled as follows:Corneal reflex: touching the pig’s cornea and checking for any movement of the eyelid (blinking).Pain reflex: pricking the inner snout of the pig with a sharp-pointed metal stick and checking for any withdrawal response.Any kicks, body convulsions or other movements were noted.Any gasping or breathing and opening/closing of the mouth were noted.The level of stun quality was graded according to criteria used to monitor unconsciousness, shown in Table 3 (extracted from Atkinson et al. [15]).

### 2.6. Statistical Analyses

Data (both behavioural and physiological) were divided into 10 s time intervals, where the first three intervals (intervals 1–3) were before the start of foam production into the box. The following 9 intervals (intervals 4–12) were with foam in the box for the two foam treatments. As the time to fill up the box differed depending on how much the pig moved inside the box, the gas pulse occurred in different intervals for different pigs. Due to measurement and observation errors, 102 (of which 24 from the Control, 24 from the Air foam and 54 from the Nitrogen foam treatment) of the total 720 intervals (60 pigs × 12 intervals) for heart rate measurements and 27 intervals (of which zero from the Control, 15 from the Air foam and 12 from the Nitrogen foam treatment) for the movement observations were excluded from the statistical analyses.

Statistical analyses were performed in SAS version 9.4 (2016; Cary, NC, USA). Descriptive statistics were calculated using Proc Means, and Proc Freq. Differences between treatments were compared within each 10 s interval and change in each variable over time (i.e., differences within variables between intervals was assessed). Heart rate was analysed as the difference between average heart rate during a baseline interval (60–30 s before the pigs were let into the stun box, i.e., when they were in the holding pen and had been there for acclimatization for one minute) and the average heart rate for the 10 s interval. Respiratory rate was not analysed due to unreliable data. Continuous variables were normally distributed (heart rate difference rate and movement (i.e., number of lines crossed in the box)) and were analysed with general linear model 1 in Proc Mixed. Binary variables (behaviours performed or not during the 10 s interval) were analysed with generalised linear model 2 in Proc Glimmix (with binomial distribution and logit link). The random effect of individual pigs was not included in Model 2 as analyses with such a model did not converge.Model 1: y = Treatment + Interval + Observation day + Sex + Interval × Treatment + PigID + eModel 2: y = Treatment + Interval + Observation day + Sex + Interval × Treatment + eWhere Treatment (Control, Air foam or Nitrogen foam), Interval (1–12), Observation day (1, 2, 3 or 4) and Sex (female or immuno-castrated male) were included as fixed effects and PigID was included as a repeated random effect (repeated over intervals).

## 3. Results

### 3.1. Filling Time of Foam in Box and Oxygen Levels

The mean time to fill the stun box with foam was 64 s (range 47–85 s) for the Air foam treatment and 86 s (range 55–137 s) for the Nitrogen foam treatment. The time to fill the box was dependent on how much the pig moved in the box during filling as the movement in itself destroyed the foam. The oxygen levels in the box after jet pulse foam destruction was 20.0% in the Air foam treatment and 0.2% in the Nitrogen foam treatment. The oxygen level in the Control treatment at the corresponding time was 19.8%.

### 3.2. Pig Behaviour and Movement

The percentages of pigs showing escape and exploring behaviours, lying and vocalisation (grunts and screams) at least once during each 10 s interval (1–12) for each treatment are shown in Figure 5, Figure 6 and Figure 7. Movement (measured as mean times the pig crossed the tape marking quadrant lines on the floor during an interval) for each treatment is shown in Figure 8.

When foam production started, the pigs in both foam treatments initially showed increased movement (Figure 8, interval 4) while aiming their explorative behaviour towards the foam (Figure 5b). A higher percentage of pigs in the foam treatments, compared to the Controls, also explored the lid, i.e., stretched their necks and heads upward towards the lid (Figure 5a, intervals 5–8). More pigs in the Control treatment explored the floor during intervals 6–12 compared to the foam treatments.

Escape attempts were observed at least once during the total treatment duration (intervals 4–12) by 15% of the pigs in the Control treatment, while the corresponding figure was 80% in both Air and Nitrogen foam treatments (Table 4). The mean number of escape attempts was 0.2 (range 0–2), 4.9 (range 0–11) and 1.9 (range 0–4) for Control, Air and Nitrogen foam treatments respectively. In the Nitrogen foam treatment, escape attempts peaked at 47% during interval 8, while in the Air foam treatment, escape attempts peaked during interval 9 at 53%. A significantly greater number of pigs in the Air foam treatment made escape attempts in intervals 10–12, i.e., when the box was completely or almost completely filled with foam.

Vocalisation by grunting was similar between treatments, with only minor differences (Figure 7a). Vocalisation by screams (Figure 7b) showed some differences between treatments. When foam production started (interval 4), a lower proportion of pigs in the foam treatments screamed compared to the controls. During intervals 7 and 8, fewer of the Nitrogen foam pigs screamed compared to the other two treatments. During intervals 10–12, a higher proportion of the Nitrogen foam pigs and Control pigs screamed compared to the Air foam treatment. Pigs were also observed screaming during convulsions after loss of posture, as described in Section 3.4 below.

Other behaviours that are also indicative of stress or respiratory distress, such as defecation and shaking, were seen at a slightly higher percentage in the foam treatments than Control, and as with escape attempts, there were little differences between Air and Nitrogen foam treatments (Table 4). For the most part, pigs in the Air foam and Nitrogen foam treatments showed similar behaviour until the Nitrogen foam pigs started to lose their posture. Starting the foam production produces a sudden noise from the pumps and gas flow, which startled 60% of the pigs. Before loss of posture, the Nitrogen foam pigs showed signs of losing their balance by swaying, sitting down or leaning against the wall. The loss of balance in combination with pigs starting to have convulsions may be the reason for the peak in movement by pigs in Nitrogen foam treatment during interval 8 (Figure 8). None of the pigs in the Nitrogen foam treatment showed gasping (Table 4).

### 3.3. Heart Rate

The mean heart rate (difference from baseline) of the pigs during intervals 1–2 for treatments Control, Air foam and Nitrogen foam is shown in Figure 9. Heart rate for pigs in the foam treatments increased after foam production started, but the difference was not statistically significant compared to the Control. The increased heart rate for pigs in Nitrogen foam peaked at approximately 20 bpm above baseline in interval 7 and then decreased sharply after interval 8.

### 3.4. Loss of Posture and Convulsions

Mean time from start of foam production to loss of posture was 57.0 s (Std ± 8.8, range 43–76 s, *n* = 20). Immediately after loss of posture, the pigs exhibited vigorous convulsions in the legs, torso, neck and/or jaw and all except one pig were vocalising (screams). The high-intensity convulsions then turned into more irregular muscle contractions including movements of the limbs and gagging (opens mouth while flexing head forward as if gasping for air). The mean time between loss of posture and the last observed muscle contraction was 132.5 s (Std ± 16.3, *n* = 20). Figure 10 shows the percentage of pigs with loss of posture and muscle contractions within each 10 s interval in relation to the lying and vocalisation (grunt and scream) behaviours.

### 3.5. Stun Quality Assessment

All pigs were assessed as level 0 according to the stun quality protocol (Table 2), i.e., considered as being in a state of deep unconsciousness. No pain or corneal reflexes, breathing or muscle contractions were observed, and the pigs’ carcasses were fully relaxed and motionless. Heartbeats could be detected in 10 pigs, while 9 had no heartbeats and one was uncertain. Thus, half of the pigs were stunned at time of sticking and bleeding, while half were most likely already dead.

## 4. Discussion

Stunning pigs with an inert gas like N_2_ is already an approved stunning method according to European Council Regulation 1099/2009; thus there is no need for additional studies to prove that this method of stunning works. To date, however, there are no published studies using only N_2_ and successfully creating a controlled stable atmosphere with high concentration N_2_ for stunning. Delivering N_2_ in foam in a closed container resulted in rapid replacement of air at high concentration N_2_ (>98%), which was a unique condition not previously reported in the context of pig stunning. This study aimed to examine animal welfare in relation to the foam itself to ensure that the foam is not highly stressful to pigs and does not cause other problems. A limitation to the study design is that it was not possible to blind the observer to the treatment allocations during the behavioural assessment, which could be a source of bias. However, since the observer did not have any stake or self-interest in the outcome, we found this risk acceptable.

In the present study, pigs showed some behavioural responses when exposed to the foam compared to only being isolated in the stun box; however, there was nothing to indicate a strong aversion to the foam. Furthermore, the initial behavioural response to air and N_2_-filled foam was similar, indicating that N_2_ itself was not initially perceived as aversive by the pigs. When foam production started, all pigs started to explore the foam with no difference between the N_2_ and air treatments. As the foam level rose in the container, most pigs were observed keeping their heads above the foam surface and exploring the lid of the box, possibly trying to avoid the foam covering their eyes and snouts. Furthermore, escape attempts increased approximately 30–40 s after the start of foam production, indicating that the foam was most aversive to pigs when starting to cover their heads. The escape attempts did not appear to be induced by panic, but instead a response to the novel experience of being covered in foam. Escape attempts may also be induced by the distress of the pig being isolated in the unfamiliar environment; however, as pigs in the Control treatment only showed a total of 0.2 escape attempts on average during intervals 1–12 and the corresponding frequency in the Air and Nitrogen foam treatments were 4.9 and 1.9, respectively, the behaviour is most likely mainly a response to the foam. There was also a higher rate of slipping in the foam treatments, which in itself could have been distressful for the pigs and possibly even caused some of the other stress-related behaviour, such as escape attempts. The slipping was caused by the wet soap on the plexiglass floor, which was necessary to enable video recording underneath the box. Transparent anti-slip tape was used to improve grip, but this wore down towards the end of the trials. Ideally, the floor should have been made of an effective anti-slip material, but for this study we had to prioritise filming from below.

Pigs were significantly more active in the two foam treatments compared to the Control during interval 4, i.e., directly after foam production started. Pigs in the Nitrogen foam treatment were significantly more active compared to both other treatments during interval 8, but this was a reflection of the loss of posture and the start of convulsions rather than behavioural movement. During loss of balance just before loss of posture and during the vigorous convulsions after loss of posture, pigs moved more between floor quadrants.

The increased heart rates seen in the N_2_ foam pigs, even though non-significant, was more likely a physiological response to exposure to hypoxic conditions than an indicator of behavioural stress; at least this has been shown in humans [16]. Electrocardiography (ECG) would have provided more detailed data on physiological response; however, the monitors are difficult to attach to the pigs and would disturb them more. With very disruptive and distressful pre-handling, training and longer acclimatisation periods might be required, but this may also interfere with normal pig behaviour during the main experiments. There is a need to develop robust reliable equipment for measuring physiological parameters of pigs that is simple to attach or, preferably, that does not require attaching a device to the pigs.

Pigs exposed to high-concentration N_2_ gas in foam showed no gasping, which is considered an indicator of the onset of breathlessness [5] and is one of the well documented aversive responses to high concentration CO_2_ gas [6,17]. These results are in line with previous studies, which have shown that exposure to inert gases or mixtures with low CO_2_ concentrations reduces gasping in pigs before loss of consciousness compared to high concentrations of CO_2_ [11,18]. Hyperventilation and rapid shaking of the head can also be indications of respiratory distress [7]. Hyperventilation was not observed; however, two pigs (10%) showed rapid head shaking when exposed to N_2_ foam, both approximately 40 s after the start of foam production. Three pigs in the Air foam treatment showed rapid head shaking, two pigs approximately 10 s after foaming started and one pig after 40 s. Therefore, it seems that rapid head shaking is likely a response to the foam rather than to respiratory distress caused by N_2_, especially since no gasping was observed. As the pigs’ behavioural responses were similar in the two foam treatments until the pigs in the Nitrogen foam treatment began to lose their posture, the pigs most likely did not experience pain or respiratory distress when exposed to high concentrations of N_2_.

Rapid induction of unconsciousness during stunning is desirable from an animal welfare point of view to minimise the period of potential distress. Exposure to 80% CO_2_ induces unconsciousness after 30–60 s [3,19]. In a previous study, stunning pigs with N_2_ showed longer times until unconsciousness compared to CO_2_ [18], but no study has used concentrations of N_2_ > 98%. With the Nitrogen foam stunning method, the minimum, mean and maximum times from start of foam production to loss of posture, which have been considered to be the first indicator of the onset of unconsciousness [6,20], were 43, 57 and 76 s respectively. As the N_2_ is contained in the foam, exposure to >98% N_2_ will occur first when the pig’s snout is covered in foam. However, the pigs destroy some foam by their movement, which means that some of the N_2_ is mixed with the air in the box, lowering its O_2_ concentration in the air above the foam. This means that lower O_2_ levels were already prevalent in the air above the foam, which likely contributed to quick stunning and LOP that sometimes even occurred before the box was filled. When LOP occurred before the box was filled, the subsequent convulsions following LOP destroyed more foam and further increased the time needed to fill the stun box. Technique development to increase foam production capacity so the box could be quickly filled with foam before LOP is experienced would likely reduce this issue and could reduce time to LOP even further.

The scientific evidence to support LOP as an indicator of unconsciousness in pigs is inconclusive [21]. Verhoeven et al. [19] showed that loss of posture was on average 10 s before electroencephalography (EEG)-based loss of consciousness in 80% and 95% CO_2_ concentrations. Both Rodriguez et al. [17] and Verhoeven et al. [19] found that muscle contractions occurred before significant changes in brain function appeared, which could possibly indicate that pigs were conscious during this period. Convulsions are not usually a cause for concern from a welfare perspective if they occur after unconsciousness, but since the onset of involuntary muscle contractions has been shown to start before, during and after the loss of consciousness in different studies, this is an issue that needs further attention in research. Studies have shown that stunning with inert gases induces more severe convulsions compared to CO_2_ stunning [11,22], but how this relates to loss of consciousness and animal welfare is not known. In the current study, all pigs showed vigorous muscle contractions (paddling, torso contractions, vocalisation) after loss of posture before shifting to more mild and irregular muscle contractions like movements of the limbs and gagging. Mean time from loss of posture to last observed muscle contraction was 131 s. Five minutes after the start of the N_2_ foam production, the pigs were assessed to be in deep unconsciousness or dead, and none of the pigs showed any signs of regaining consciousness during bleeding. Longer time to unconsciousness may negatively affect meat and carcass quality [11], and severe convulsions can also be an issue in relation to these aspects.

Based on the results found in this study, stunning using foam to deliver high-concentration N_2_ may be a viable alternative to high-concentration CO_2_ stunning and might lead to improved animal welfare. Nitrogen makes up 78% of atmospheric air, and it is rather economical to extract the gas from air, which makes it a convenient and economically competitive alternative to CO_2_. However, further studies are needed to assess the stunning of slaughter-weight pigs, investigate the effects of group stunning (two or more pigs) and determine the adequate time needed in anoxic conditions after stunning to ensure unconsciousness throughout the subsequent killing procedures at slaughter. Further development of the method to increase foam generation capacity for quicker filling times will also likely decrease stunning times, and there is also a need to be able to handle groups of slaughter-ready pigs. For the industry, one important issue is also how N_2_ foam stunning affects meat and carcass quality, especially in comparison to carbon dioxide stunning and electrical stunning, which are the current norms for pigs at slaughter. Another issue that needs further attention is the foaming agent, which is introduced to the slaughter process, to ensure that foam residues will not be of concern from a food safety perspective.

## 5. Conclusions

This study shows a proof of concept for using N_2_-filled foam for stunning of pigs. The main conclusions are:The Anoxia system using N_2_-filled foam was effective at quickly purging the air from the box to create and maintain stable anoxic conditions.The pigs showed moderate aversive behaviours to foam. They explored the foam with their snout when the foam generator started; however, they seemed to avoid putting their heads and snouts into the foam when foam levels became high, at which point escape attempts through the lid also increased. The escape attempts did not appear to be induced by panic, but instead a response to the novel experience of being covered in foam.The pigs did not show any aversive behaviours to the N_2_-filled foam compared to the air-filled foam. Physiological responses, such as increased heart rate, were found, but this was expected.Further studies should be made testing the N_2_ foam technique stunning fattening pigs at slaughter age and small groups together.

## Figures and Tables

**Figure 1 animals-10-02210-f001:**
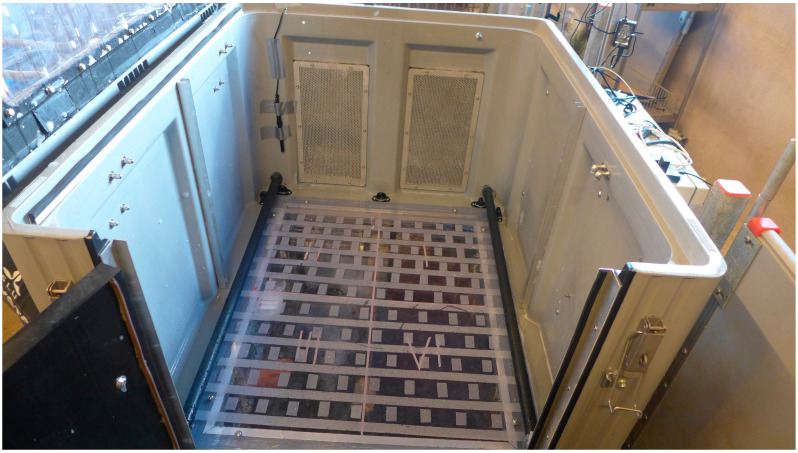
The stun box interior with transparent floor and anti-slip tape. The two flat rectangular foam generators are seen on the far side and the black jet pulse system along the base of the long sides.

**Figure 2 animals-10-02210-f002:**
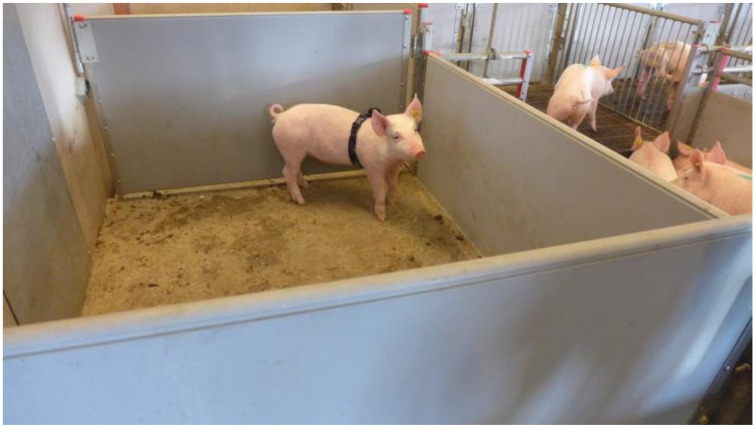
The holding pen, next to the group pen, with a pig wearing the Bioharness (black chest band). Baseline heart rate was measured during the end of the acclimatisation period of two minutes.

**Figure 3 animals-10-02210-f003:**
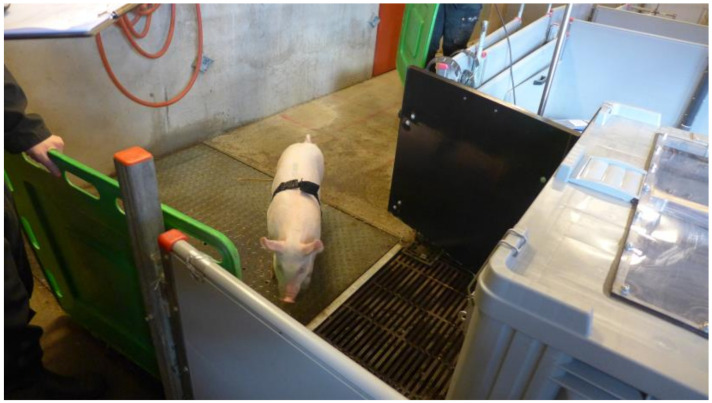
A pig being moved from the holding pen to the Anoxia stun box.

**Figure 4 animals-10-02210-f004:**
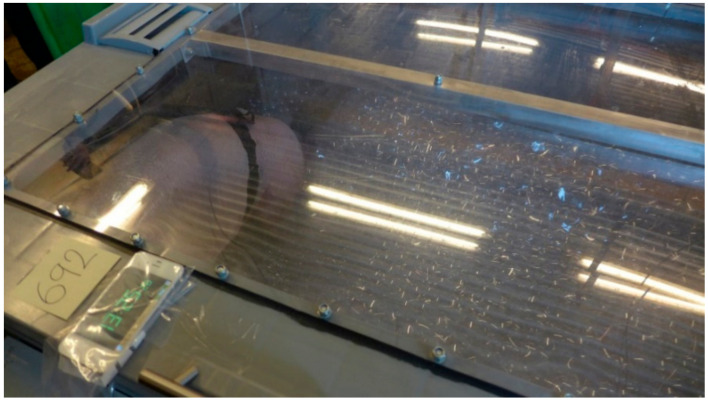
A pig inside the stun box, exploring the floor and foam during a foam treatment.

**Figure 5 animals-10-02210-f005:**
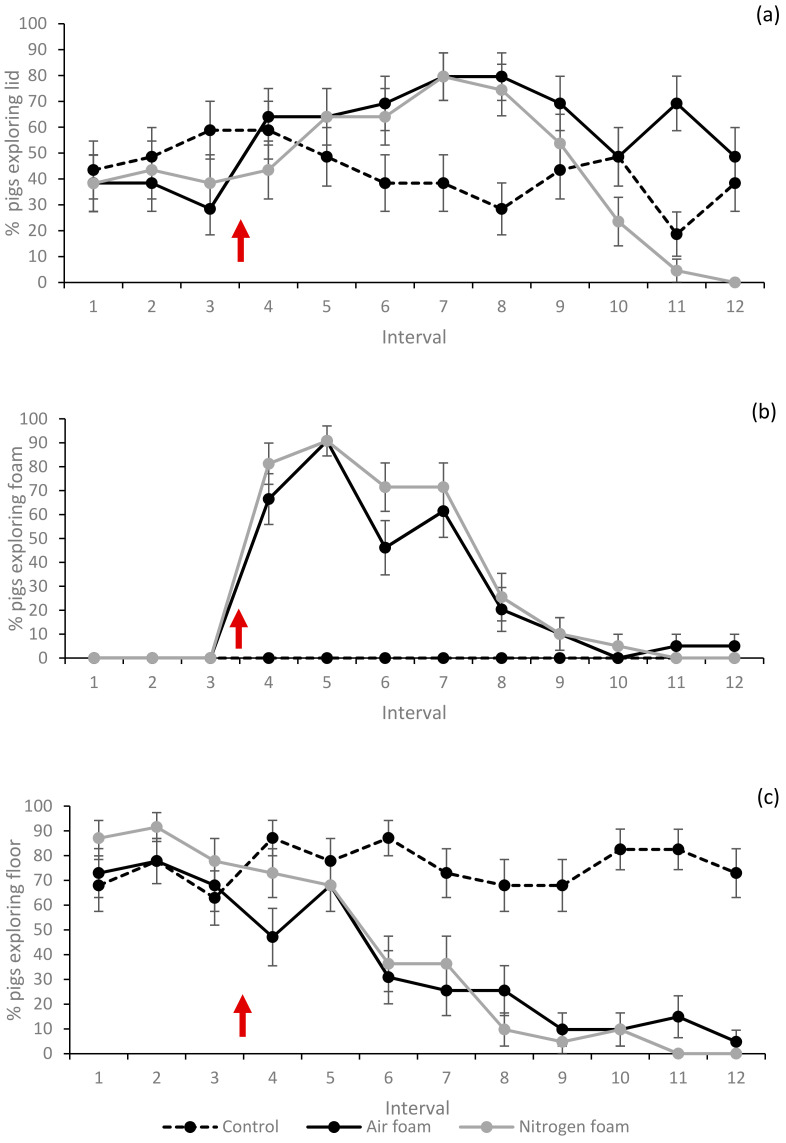
Percentage (%) of pigs exploring the (**a**) lid, (**b**) foam and (**c**) floor at least once during the interval for the three treatments (LSmeans and standard error, *N* = 60). Each interval (1–12) represents 10 s. Red arrow indicates start of foam production. Significant difference (*p* < 0.05) when standard error bars do not overlap (both between treatments within interval and between intervals within treatment).

**Figure 6 animals-10-02210-f006:**
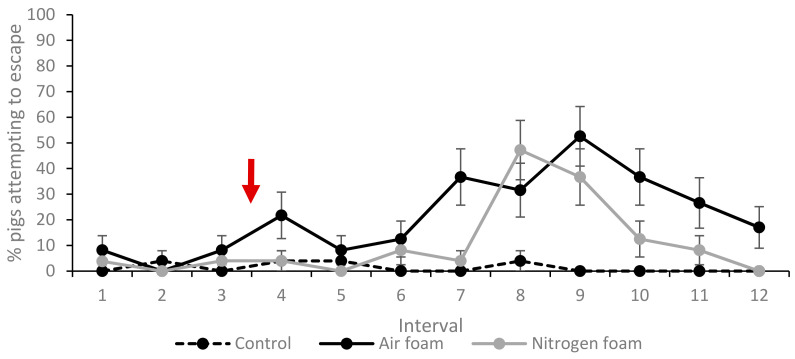
Percentage (%) of pigs attempting to escape at least once during the interval for the three treatments (LSmeans and standard error, *N* = 60). Each interval (1–12) represents 10 s. Red arrow indicates start of foam production. Significant difference (*p* < 0.05) when standard error bars do not overlap (both between treatments within interval and between intervals within treatment).

**Figure 7 animals-10-02210-f007:**
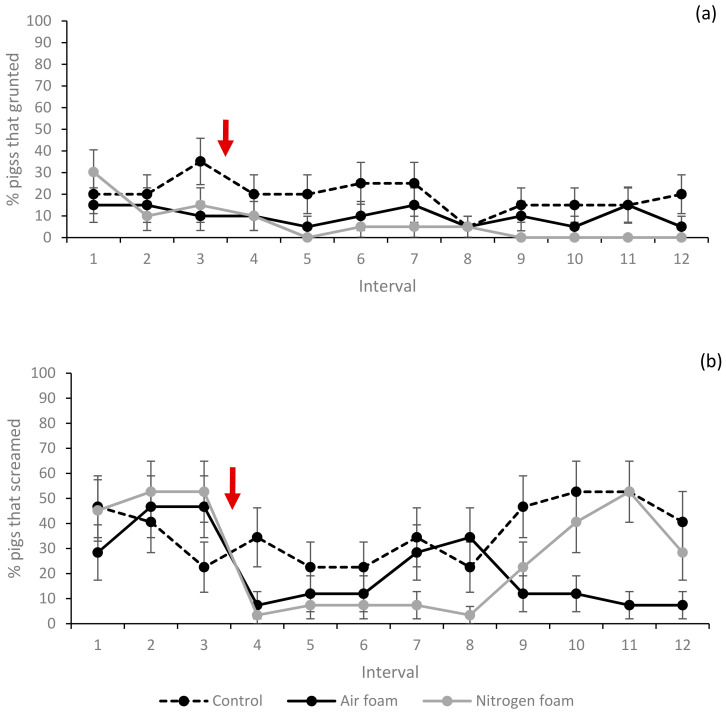
Percentage (%) of pigs vocalising ((**a**) grunt (**b**) scream) during the interval for the three treatments (LSmeans and standard error, *N* = 60). Each interval (1–2) represents 10 s. Red arrow indicates start of foam production. Significant difference (*p* < 0.05) when standard error bars do not overlap (both between treatments within interval and between intervals within treatment).

**Figure 8 animals-10-02210-f008:**
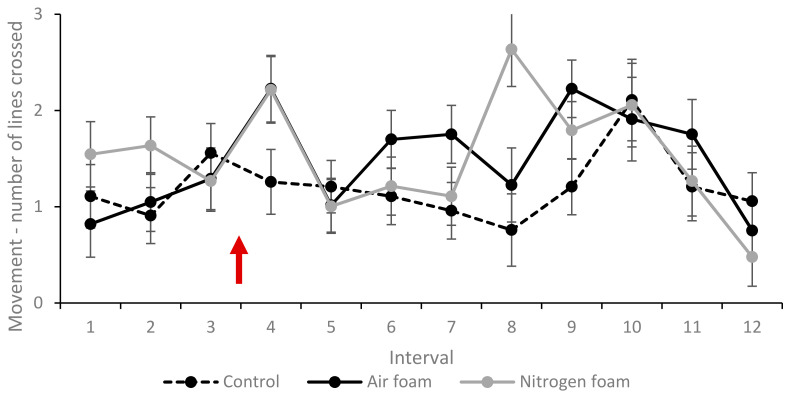
Movement within the box described by number of quadrant lines the pig crossed during an interval (LSmeans and standard error, *N* = 60). Each interval (1–12) represents 10 s. Red arrow indicates start of foam production. Significant difference (*p* < 0.05) when standard error bars do not overlap (both between treatments within interval and between intervals within treatment).

**Figure 9 animals-10-02210-f009:**
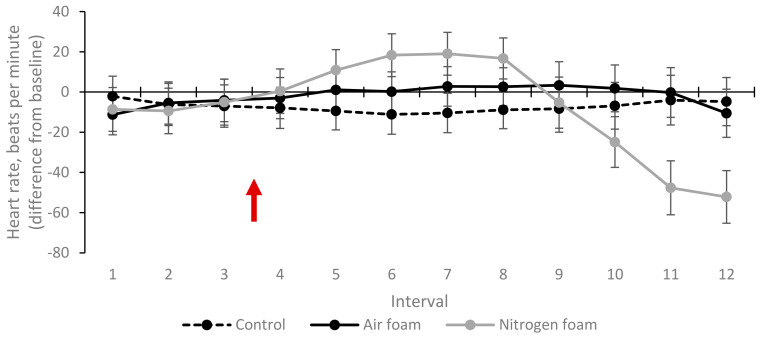
Heart rate (difference from baseline) for the three treatments (LSmeans and standard error, N = 60). Each interval (1–12) represents 10 s. Red arrow indicates start of foam production. Significant difference (*p* < 0.05) when standard error bars do not overlap (both between treatments within interval and between intervals within treatment).

**Figure 10 animals-10-02210-f010:**
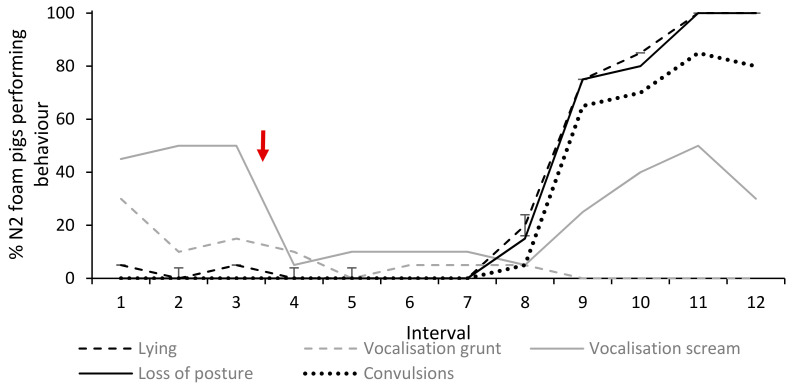
Percentage (%) of pigs in the Nitrogen foam treatment (*n* = 20) showing lying and vocalisation behaviours in relation to time of loss of posture and onset of muscle contractions. Each interval (1–12) represents 10 s. Red arrow indicates start of foam production.

**Table 1 animals-10-02210-t001:** Mean value and standard deviation of pig weights within the treatments Control, Air foam and Nitrogen foam.

Treatment	Control (*n* = 20)		Air Foam (*n* = 20)		Nitrogen Foam (*n* = 20)		Total (*N* = 60)	
	Mean	Std	Mean	Std	Mean	Std	Mean	Std
Weight (kg)	28.0	3.4	27.4	3.6	27.9	3.3	27.8	4.4

**Table 2 animals-10-02210-t002:** Pig behaviours and their definitions and registration.

Behaviour	Definition	Registration
Defecation	Self-explained	% of pigs showing the behaviour during treatments
Escape attempt	Kicks with front or hind legs, jumps up or pushes lid, door or wall	No. of times observed within each 10 s interval
Exploring floor	Snout touches the floor or the jet pulse tubes	No. of times observed within each 10 s interval
Exploring foam	Snout touches foam, pig is actively seeking contact with foam	No. of times observed within each 10 s interval
Exploring lid	Head stretches upward with snout pointing toward the lid	No. of times observed within each 10 s interval
Exploring wall	Snout touches the wall	No. of times observed within each 10 s interval
Gasping	Deep breath through a wide-open mouth, which may involve stretching of the neck	% of pigs showing the behaviour during treatments
Lying	Lying position with side of body or belly in contact with the floor	No. of times observed within each 10 s interval
Movement	Both front feet moved from one floor-square (marked with tape) to another	No. of times observed within each 10 s interval
Shaking	Shakes body	% of pigs showing the behaviour during treatments
Sitting	A sitting, upright position with the hind legs resting on the ground and front legs straight with hooves on the floor	% pigs showing the behaviour during treatments
Slipping	One or more hooves sliding quickly and uncontrolled on floor	% of pigs showing the behaviour during treatments
Startle	A sudden, involuntary movement/jump in response to unexpected sound or movement	% of pigs showing the behaviour during treatments
Vocalisation	Score 1 = occasional or repeated gruntsScore 2 = occasional or frequent screams	Behaviour occurring or not within each 10 s, one score per 10 s interval

**Table 3 animals-10-02210-t003:** Stun quality protocol (from Atkinson et al. [15]).

Stun Quality Level	Description of Signs
0	If a pig showed no reflexes or signs mentioned below, it was considered as being in a state of deep unconsciousness and posed no risk for poor animal welfare.
1	If a pig showed kicks or other movements or infrequent gasps but had no eye or pain reflexes when checked, it was considered adequately stunned but justified continual monitoring.
2	If a pig displayed frequent gasps (opening and closing of the mouth with or without stretching of neck), kicks or body convulsions, but was found to have no eye or pain reflexes, it was re-stunned as a precaution to avoid recovery. The stun depth was considered unacceptable due to the risk that the animal could recover.
3	If a pig showed corneal or cilia blink reflex at sticking, with or without kicking or convulsions, it would be immediately re-stunned and the recovery risk was thought to be imminent and the stun was considered inadequate.
4	If a pig showed spontaneous blinking, righting reflex, vocalisation or pain reflex, it was considered as indicating some form of consciousness and a high risk for poor welfare and the stun was considered inadequate.

**Table 4 animals-10-02210-t004:** Percentage (%) of pigs showing the behaviour at least once during treatments (i.e., intervals 4–12).

Behaviour	Control	Air Foam	Nitrogen Foam
	(*n* = 20 pigs)	(*n* = 20 pigs)	(*n* = 20 pigs)
Defecation	5%	25%	20%
Escape attempt	15%	80%	80%
Exploring floor	95%	80%	80%
Exploring foam	0%	100%	100%
Exploring lid	85%	95%	90%
Exploring wall	85%	40%	30%
Gasping	0%	0%	0%
Lying	20%	45%	100%
Shaking	0%	15%	10%
Sitting	10%	40%	60%
Slipping	0%	45%	20%
Startle	10%	60%	60%
Vocalisation grunt	75%	50%	20%
Vocalisation scream	80%	45%	95%

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
