# Peer review of "Responses of Pigs to Stunning with Nitrogen Filled High-Expansion Foam"

_animals, 2020, doi:10.3390/ani10122210_

Round 1
Reviewer 1 Report
A very interesting and important work to improve stunning methods in pigs. I just have 2 minor criticisms and a suggestion for further study. My first criticism is a consequence for the richness of the results. Readers get a little bit lost in the middle of the data. It would make sense to have the results presented by more "functional" groups. I mean something like indicators of stress (those that are well-known and the putative ones), indicators related to reversibility, indicators related to "preserved cardiac function... It is both a comment and a suggestion.
My second criticism is also related to the "Results" section. On lines 341 "Stun quality assessment", you describe again details that are part of the Material and Methods. Please remove it and just edit your Mat and Meths section to be sure that all necessary details are well described. But the Results about "Stun quality" would probably gain more interest in being more detailed.
My suggestion for further research is also something that you should, in my opinion, discuss in the Discussion section. It is related to using Heart rate Variability analysis. This method could be very interesting and easy to perform, using your system. It would provide valuable information about the level of stress triggered by the foam and is much practical compared to EEG. At least, it would be great to explain why you reject those possible indicators.
Author Response
A very interesting and important work to improve stunning methods in pigs. I just have 2 minor criticisms and a suggestion for further study.
We thank the reviewer for these comments.
My first criticism is a consequence for the richness of the results. Readers get a little bit lost in the middle of the data. It would make sense to have the results presented by more "functional" groups. I mean something like indicators of stress (those that are well-known and the putative ones), indicators related to reversibility, indicators related to "preserved cardiac function... It is both a comment and a suggestion.
This is a worthy point by the reviewer, and we are aware there are many different ways possible to present the data. We have thoroughly discussed the outline of the Results section within our group and considered different alternatives, including the suggestion by the reviewer to reorganise the results in functional groups. However, we do not feel this will greatly enhance the readers understanding. We decided that presenting the results in chronological order is the most logic outline and aids the reader to find results in as they happened. In the Discussion section, we then explain the results in relation to “different functions” such as indicators of stress etc.
My second criticism is also related to the "Results" section. On lines 341 "Stun quality assessment", you describe again details that are part of the Material and Methods. Please remove it and just edit your Mat and Meths section to be sure that all necessary details are well described. But the Results about "Stun quality" would probably gain more interest in being more detailed.
The first sentence under “Stun quality assessment” in the Results section was moved to the Materials and methods section (L 197-198).
My suggestion for further research is also something that you should, in my opinion, discuss in the Discussion section. It is related to using Heart rate Variability analysis. This method could be very interesting and easy to perform, using your system. It would provide valuable information about the level of stress triggered by the foam and is much practical compared to EEG. At least, it would be great to explain why you reject those possible indicators.
We agree with the reviewer that heart rate variability would be interesting. However, the method to measure heart rate used in the current study, the Zephyr BioHarness, was not reliable enough based on our experiences. The advantages with the BioHarness is that it measures both heart rate, respiratory rate and activity in the same devise, which means data is synchronised, and it is easy to attach on the pig and does not seem to disturb the pig once attached. However, there were some difficulties in ensuring a good data quality, especially for small sized pigs. The equipment is designed for humans to be used during training, thus it is not optimal for application on pigs during stunning. To be able to get reliable data, an ECG monitor is most likely the best option, but as it is more difficult to attach on the pig and will disturb the pig more, it probably demands repeated training and acclimatization to not interfere the pig too much during experiments. We have failed to find a robust reliable measuring equipment of physiological parameters on pigs, preferably without having to attach a devise to the body of the pig, thus this is an identified need for development.
We have added a short paragraph to the discussion to address this issue, see L399-407.
Reviewer 2 Report
This is an important study, especially in a time when disease outbreaks are increasingly likely, and rapid and humane methods of slaughter are needed. These results will be of interest to many readers working in governmental and agricultural industries.
The study is clearly written and generally well described. It was fairly straightforward to follow. I have one major concern regarding video scoring; otherwise, I have mostly minor comments.
L 61-62 I do not think that this result from rats is relevant here. Rats are burrowing rodents with very high sensitivity to oxygen levels in the atmosphere. They show a strong aversion to both argon- and nitrogen-induced hypoxia. Therefore, I do not believe that rats tolerating nitrogen a bit more than argon tells us much about pigs’ reaction to either gas.
L 148 Heart and breathing rate were recorded
L 177 Check the sentence here.
L 177 I imagine the observer could not be blinded to treatments, but what steps were taken to avoid bias? Was the observer perhaps naïve to the study aim and hypothesis? Some measures must be used for this study to be publishable. Including another independent observer and giving inter-observer reliability would also be useful.
L 184-185 I think I misunderstand here. In the control treatment, would starting behavioural observation 30 s before the end of the acclimatization period not be the equivalent of the observation period for foam treatments? In lines 154 and 158 you say that foam production started after the acclimatization period. In line 147 you say pigs were acclimatized in the stun box for 2 min before the treatment started. Therefore, if behavioural observations started 30 seconds before foam production started (line 184), I understand that observations started during the last 30 seconds of acclimatization in the stun box. Should observations in the control treatment not also start during the last 30 s of the acclimatization period, rather than after the acclimatization period?
L 217 This should read 60 pigs x 12 intervals. Also, can you state how many intervals were excluded from each treatment?
L 265 Given that the range of time it took for the box to fill with air foam was 47-85 s, at least one pig still had foam in intervals 10-11, correct? And for some pigs the foam was already destroyed before intervals 10-12, correct?
Figures 9 and 10 The red arrow does not appear to be in the correct location.
L 351 Perhaps use the word ‘method’ rather than ‘principal’
L 371-373 I do not see the relevance of this statement. Yes, pigs may have shown escape attempts because they saw the Plexiglas lid as a potential way out. If the lid had been opaque, they may have not shown this escape behaviour, but their motivation to escape would have been the same (they would just have no reason to express it). The aim is not to eliminate behaviour, but the motivation to perform the behaviour. This statement is only relevant if you say that slipping was caused by escape attempts – in this case, recommending an opaque lid makes sense to avoid injury and distress from slipping.
L 447 What would you consider “strong” aversive reaction to foam? Pigs exposed to foam had more defecation, escape attempts, and startle. The foam clearly induced anxiety. On what basis do you say that their aversion wasn’t “strong”? Based on your observations of these pigs, you may conclude that aversion wasn’t strong, but some discussion about why you consider aversion weak or moderate versus strong is warranted. Did you have pre-determined criteria for intensity of aversion? It may be more accurate or objective to conclude that pigs showed aversion to the foam itself, although you believe this aversion was not strong (but you need to explain why in the Discussion).
L 449-450 Again, I do not see the relevance of this statement, unless you say that this would help avoid slipping. Otherwise, it is not related to welfare.
Additional comments:
Since only about 50% of the pigs had no heartbeat after 5 min, what does this mean for using this for slaughter? Quickly bleeding the pigs before they regain consciousness may not be feasible if several animals are killed at once. Can you recommend that the time exposed to foam should likely be longer than 5 min, and that the exact time should be determined in future studies? You allude to this in your conclusion, but a discussion of this in the Discussion is warranted.
Slipping is a welfare concern; it could cause injury, but it is also likely distressing to the pigs. Can you explain more about the context of slipping? Was it caused by the foam per se (wet) or by escape attempts? Can you recommend using different flooring substrate to minimize slipping?
Author Response
This is an important study, especially in a time when disease outbreaks are increasingly likely, and rapid and humane methods of slaughter are needed. These results will be of interest to many readers working in governmental and agricultural industries.
The study is clearly written and generally well described. It was fairly straightforward to follow. I have one major concern regarding video scoring; otherwise, I have mostly minor comments.
We thank the reviewer for these comments.
L 61-62 I do not think that this result from rats is relevant here. Rats are burrowing rodents with very high sensitivity to oxygen levels in the atmosphere. They show a strong aversion to both argon- and nitrogen-induced hypoxia. Therefore, I do not believe that rats tolerating nitrogen a bit more than argon tells us much about pigs’ reaction to either gas.
Thanks for pointing this out and we agree so the sentence about rats’ response to nitrogen has been removed together with the reference.
L 148 Heart and breathing rate were recorded
Thanks for seeing this, we changed was to were in L 148.
L 177 Check the sentence here.
The sentence has been rephrased in L 177-178
L 177 I imagine the observer could not be blinded to treatments, but what steps were taken to avoid bias? Was the observer perhaps naïve to the study aim and hypothesis? Some measures must be used for this study to be publishable. Including another independent observer and giving inter-observer reliability would also be useful.
We do see the point made by the reviewer, but for this type of applied study we did not have the economic resources to have multiple independent observers analyse the video recordings as this was a very time consuming and demanding analysis. Furthermore, it was clearly not possible to blind the observer to the treatments. This is a limitation to the study, but by describing the method and clearly stating that the video recordings were analysed by one observer, we believe that it gives enough information to allow the reader to evaluate the credibility of the results and the risk of bias. The person who conducted the behavioural observations had no self-interest in the outcome of the study or any conflict of interest.
L 184-185 I think I misunderstand here. In the control treatment, would starting behavioural observation 30 s before the end of the acclimatization period not be the equivalent of the observation period for foam treatments? In lines 154 and 158 you say that foam production started after the acclimatization period. In line 147 you say pigs were acclimatized in the stun box for 2 min before the treatment started. Therefore, if behavioural observations started 30 seconds before foam production started (line 184), I understand that observations started during the last 30 seconds of acclimatization in the stun box. Should observations in the control treatment not also start during the last 30 s of the acclimatization period, rather than after the acclimatization period?
We understand the confusion, as the description of the observation period in the Control treatment was not well explained in the manuscript. The observations of the pigs in the control group also started during the last 30 seconds of the acclimatization period, thus the period is equivalent to the period for the foam treatments. The sentence describing the observations of the control group has been changed to clarify this (L 15-186).
L 217 This should read 60 pigs x 12 intervals. Also, can you state how many intervals were excluded from each treatment?
Thanks for pointing this out as it was not clear. The number was changed to 60 pigs x 12 intervals and we have added information on the number of intervals that were excluded from each treatment (L 218-221).
L 265 Given that the range of time it took for the box to fill with air foam was 47-85 s, at least one pig still had foam in intervals 10-11, correct? And for some pigs the foam was already destroyed before intervals 10-12, correct?
Yes, a majority of the pigs in the Air foam treatment were in high levels of foam during intervals 10-11. The sentence has been rewritten to make this clearer (L 269-270).
Figures 9 and 10 The red arrow does not appear to be in the correct location.
Thanks for noticing this, we have adjusted the arrows and double and triple checked all figures.
L 351 Perhaps use the word ‘method’ rather than ‘principal’
The word has been changed from principal to method (L 361).
L 371-373 I do not see the relevance of this statement. Yes, pigs may have shown escape attempts because they saw the Plexiglas lid as a potential way out. If the lid had been opaque, they may have not shown this escape behaviour, but their motivation to escape would have been the same (they would just have no reason to express it). The aim is not to eliminate behaviour, but the motivation to perform the behaviour. This statement is only relevant if you say that slipping was caused by escape attempts – in this case, recommending an opaque lid makes sense to avoid injury and distress from slipping.
We agree with the reviewer and decided to remove this reasoning from the manuscript.
L 447 What would you consider “strong” aversive reaction to foam? Pigs exposed to foam had more defecation, escape attempts, and startle. The foam clearly induced anxiety. On what basis do you say that their aversion wasn’t “strong”? Based on your observations of these pigs, you may conclude that aversion wasn’t strong, but some discussion about why you consider aversion weak or moderate versus strong is warranted. Did you have pre-determined criteria for intensity of aversion? It may be more accurate or objective to conclude that pigs showed aversion to the foam itself, although you believe this aversion was not strong (but you need to explain why in the Discussion).
This is a very good comment. We changed the description to moderate aversive behaviours and tried to describe the reasoning behind this assessment. (L 474-481)
L 449-450 Again, I do not see the relevance of this statement, unless you say that this would help avoid slipping. Otherwise, it is not related to welfare.
We agree with the reviewer and decided to remove this reasoning from the manuscript.
Additional comments:
Since only about 50% of the pigs had no heartbeat after 5 min, what does this mean for using this for slaughter? Quickly bleeding the pigs before they regain consciousness may not be feasible if several animals are killed at once. Can you recommend that the time exposed to foam should likely be longer than 5 min, and that the exact time should be determined in future studies? You allude to this in your conclusion, but a discussion of this in the Discussion is warranted.
We agree that the exposure time to the anoxic atmosphere is very important when applied in commercial slaughter, to ensure pigs are dead or irreversibly stunned before sticking and bleeding. However, the results from this study cannot be used as a basis for such recommendations on exposure times. This need for future studies to determine exposure times is already included in the Discussion section of the manuscript, L 457-460.
Slipping is a welfare concern; it could cause injury, but it is also likely distressing to the pigs. Can you explain more about the context of slipping? Was it caused by the foam per se (wet) or by escape attempts? Can you recommend using different flooring substrate to minimize slipping?
The foam made the floor wet and slippery, and as the test box was equipped with a transparent floor to enable video recordings from underneath the box, the floor did not have a non-slip surface. A transparent non-slip tape was attached to the floor to decrease slipperiness, but the tape was worn down and was less effective by the end of the trials. A stunning box for commercial use would not need the transparent floor, and with a better surface on the floor, slipping could be prevented. We have added a short section about slipping to the Discussion (L 381-387).